# Properdin Is a Modulator of Tumour Immunity in a Syngeneic Mouse Melanoma Model

**DOI:** 10.3390/medicina57020085

**Published:** 2021-01-21

**Authors:** Izzat A. M. Al-Rayahi, Lee R. Machado, Cordula M. Stover

**Affiliations:** 1Department of Respiratory Sciences, University of Leicester, Leicester LE1 9HN, UK; izzatalrayahi@gmail.com (I.A.M.A.-R.); cms13@le.ac.uk (C.M.S.); 2Department of Medical Laboratory Technology, College of Health and Medical Technology, Middle Technical University, Baghdad 10047, Iraq; 3Centre for Physical Activity and Life Sciences, University of Northampton, Northampton NN1 5PH, UK

**Keywords:** CCL2, MDSC, melanoma, B16

## Abstract

*Background and Objectives*: Tumours are often low immunogenic. The role of complement, an innate immune defence system, in tumour control has begun to be elucidated, but findings are conflicting. A role for properdin, an amplifier of complement activation, in tumour control has recently been implicated. *Materials and Methods*: Properdin-deficient and congenic wildtype mice were injected subcutaneously with B16F10 melanoma cells. Tumour mass and chemokine profile were assessed. The frequencies of CD45^+^CD11b^+^ Gr-1^+^ cells were determined from tumours and spleens, and CD206^+^ F4/80^+^ cells were evaluated in spleens. Sera were analysed for C5a, sC5b-9, and CCL2. *Results*: Whilst there was no difference in tumour growth at study endpoint, properdin-deficient mice had significantly fewer myeloid-derived suppressor cells (MDSCs) in their tumours and spleens. Splenic M2 type macrophages and serum levels of C5a, sC5b-9, and CCL2 were decreased in properdin-deficient compared to wildtype mice. *Conclusions*: The presence of intact complement amplification sustains an environment that lessens potential anti-tumour responses.

## 1. Introduction

Mouse models of various tumours have been employed as preclinical models to inform or direct management of human disease, though they have their limitations [1]. Syngeneic tumour models involve implantation of tumour cells of the same genetic background as the host and therefore mimic the human situation when tumours arise in an immunocompetent host after having escaped immune recognition [2]. B16 melanoma models are poorly immunogenic [3]. In comparison to highly immunogenic solid tumour models, B16-induced tumours show relatively greater expansion and a paucity in expression of genes characteristic of activated T cells and dendritic cells. Immune suppressive cell populations (regulatory T cells and myeloid-derived suppressor cells (MDSCs)) are detectable in tumour as well as spleen and associate with a poor response to experimental immunotherapy [4]. The mouse spleen is able to receive progenitor cells from bone marrow (in the adult) and, unlike human, provides a suitable niche for hematopoietic differentiation [5], from where they may seed target tissues including tumours [6]. Monocytic replenishment from spleen in a mouse model of lung adenocarcinoma was CCR2 dependent and impacted on tumour growth [7].

Recent studies have highlighted the importance of innate immune pathways in modulating the tumour microenvironment. Specifically, complement anaphylatoxins lead to activation, movement, and proliferation of cells. C5a and C3a bind to MDSCs, tumour infiltrating lymphocytes, and tumour cells, and thereby dampen adaptive immunity and consequently increase metastatic potential [8]. In a syngeneic model of cervical cancer, primary tumours in the flanks showed C3 dependent tumour growth, activation of the classical pathway of complement, and C5a mediated recruitment of CD45^+^ CD11b^+hi^ Gr1^+^ MDSC into tumours [9].

B16F10 is a melanoma cell line which is injected subcutaneously and allows investigation of an orthotopic tumour (in the skin). A 14-day model of B16F10 melanoma growth using wildtype and C3^−/−^ mice showed that smaller tumour development was associated with greater IL-10 expression in C3^−/−^ CD8^+^ T cells, which were increased, while the numbers of CD45^+^ CD11b^+^ Gr1^+^ tumour infiltrating leukocytes remained comparable [10].

Similarly, a C3aR-positive host environment was found to benefit tumour growth in another 14-day model of B16F10 melanoma comparing wildtype and C3aR^−/−^ mice. C3aR^−/−^ mice showed a change in the proportions of different leukocyte subpopulations in tumour infiltrates and had increased CCL5 in lysates from tumours. Daily intraperitoneal injections with C5aR or C3aR antagonists of tumour-bearing mice led to a significant reduction in the tumour mass compared to the vehicle controls [11]. In contrast, a dependence on C3 or C5 could not be demonstrated in a different study of B16F10 melanoma growth over 14 days but showed a significant contribution of C1q to tumour development [12]. These differences may point to differences in cell culture conditions or in a subline that has evolved, particularly as metastatic potential was evaluated in the latter but not the two former studies.

Properdin is a positive regulator of the alternative pathway of complement activation that stabilises the alternative pathway convertases which generate C3a and C5a. We have previously studied the in vitro effect of B16F10 conditioned medium on the profile of bone marrow-derived macrophages of properdin-deficient and wildtype mice. We found that macrophages from properdin-deficient mice were skewed towards an M2 phenotype and speculated that “properdin insufficiency may promote a tumour environment that helps the tumour evade the immune response” [13].

Our aim was to determine whether properdin is an important modulator of tumour growth in vivo. We found that while overall tumour growth was not significantly impacted by the absence of properdin, there was significantly less CCL2 chemokine and fewer Gr1^+^ CD11b^+^ MDSCs as well as M2 type macrophages compared with wildtype counterparts.

## 2. Materials and Methods

### 2.1. Mice

All animal experiments were carried out after acquiring approval from the institutional ethical review committee (University of Leicester) and conducted in accordance with United Kingdom Home Office regulations (project license PPL 80/2354—approved 1 December 2014). The severity of the protocol used in this paper was classed as moderate. Age-matched male congenic mice were taken from the properdin-deficient mouse colony held at the University of Leicester [14]. Wildtype and properdin-deficient mice were subcutaneously injected with tumour cell suspensions in their flanks and monitored for their wellbeing and tumour growth every other day to daily. Some received an additional injection of luciferin where indicated (imaging). Mice reached their endpoint (tumour size: 1 cm in one dimension) within 14 days.

### 2.2. Cells

Mouse melanoma cells B16F10 were grown as described [13] and extensively washed in phosphate buffered saline (PBS) prior to injection at 1.6 × 10^5^ cells/100 μL. B16F10-luc cells (mouse melanoma cells expressing luciferase) were kindly provided by Dr. Victoria Brentville (Scancell, Nottingham, UK) and injected at a dose of 4 × 10^5^ cells/100 μL. Cells were confirmed to be mycoplasma negative using the EZ-PCR Mycoplasma Test Kit (Biological Industries, Cromwell, CT, USA).

### 2.3. Determination of Tumour Mass, Bioimaging

Callipers were used to measure baseline tumour width and length (mm^3^). The following formula was used to calculate tumour volume because of the ellipsoid shape of the tumour mass: 0.5 × A × B^2^, where A and B are the diameters in mm. In other groups, tumours were weighed after removal at necropsy. Bioluminescence was used in others. To do this, mice which had received B16F10-luc cells were injected in the scruff of their neck with 150 μg/kg luciferin subcutaneously and their tumours were imaged using IVIS^®^ Spectrum In Vivo Imaging System (Caliper Life Sciences, Hopkinton, MA, USA) a pilot experiment to discern distribution kinetics of luciferin. Light emission was measured under anaesthesia with recovery 10 min after substrate injection.

### 2.4. Proteome Array Analysis

Tumour lysates were prepared and membranes (Proteome Profiler™ Array; R&D Systems, Abingdon, UK) were hybridised with 300 µg total protein. Pixel densities on developed X-ray film were analysed using a transmission mode scanner and image analysis software. A template was created to analyse the pixel density of each spot on the array. Then signal values were exported to an Excel (Microsoft Excel 2013) spreadsheet for analysis. The average signal of the pair of duplicate spots representing each cytokine was determined. The signal from a clear area of the array or negative control spots was used as a background value, and the background signal was subtracted from each spot.

### 2.5. ELISA

ELISA sandwich assays were used to detect C5a (R&D Systems, DY2150), CCL2 (Affymetrix eBioscience, 88-7391, Fisher Scientific Loughborough, UK), and C5b-9 (MyBioSource, MBS703522, San Diego, CA, USA) according to the manufacturers’ instructions.

### 2.6. Flow Cytometry

For flow cytometry, 1 × 10^6^ cell/mL were resuspended in PBS supplemented with 1% FBS. The cells were incubated with purified anti-Fc receptor blocking antibody (anti-CD16/CD32 from Biolegend, San Diego, CA, UK) for 30 min in ice before staining with saturating concentrations of antibodies specific for CD45 (APC, clone RA3-6B2; eBioscience), CD11b (PE, clone M1/70; Ebioscience), Gr-1 (FITC, clone RB6-8C5; eBioscience), F4/80 (FITC, clone BM8, 1:250; Biolegend), or CD206 (APC, clone C068C2, 1:250; Biolegend). Cells were washed and analysed using either CellQuest or FACSDiva 6.0 software (BD Biosciences, San Jose, CA, USA).

### 2.7. Statistical Analysis

Statistical analysis was performed using GraphPad v8.0.1 (GraphPad Software) using unpaired *t*-test. Serum and single cell suspensions were analysed from three independent experiments. ELISA for CCL2, C5b-9, and C5a in sera and flow cytometry results for cell phenotyping were performed on identical mice.

## 3. Results

### 3.1. Properdin-Deficient and Wildtype Mice Have Comparable Tumour Load

A pilot study determined subcutaneous, non-infiltrative, and non-metastasising growth. The tumour was non-irritant and innocuous. The endpoint (diameter of tumour 1 cm) was reached between 12 and 14 days. Subsequently, properdin-deficient and congenic wildtype mice were implanted subcutaneously with the B16F10 melanoma cell line. Tumours were measured by callipers and the volume was calculated using the formula for ellipsoid shapes (Figure 1a). As tumours were often uneven in their circumferences, these measurements approximated the real mass. Therefore, tumours were excised and weighed (Figure 1b). In both cases, there was a bimodal distribution, with 3/11 tumours being much larger than the rest and 5/14 wildtype mice clearly having large tumour weights. Properdin-deficient mice had much less variable growth of tumour mass. Statistically, there was no difference between the groups. For in vivo assessment of the tumour burden, we used the B16-Luc line which expresses luciferase. The tumour mass was captured by bioluminescent imaging (Figure 1c). The distribution in average radiance was widespread; there was no significant difference in derived tumour mass between wild type and properdin-deficient mice. One caveat is that necrotic areas alter luminescence intensity, so tumour mass seemed to be the more reliable measurement.

### 3.2. Inflammatory Marker Expression in B16 Tumours is Altered in Properdin-Deficient Mice

Tumours extracted from a subgroup of mice matched for the size of tumours were analysed quantitatively for the presence of important mediators. Of forty mediators evaluated, only TIMP1, IL-13, CCL2, and CCL3 were significantly different between the tumour lysates prepared from the two genotypes, where properdin-deficient mice showed reductions in the chemoattractants CCL2, CCL3, the Th2 cytokine IL-13, and the metalloproteinase inhibitor TIMP-1 (Table 1 and Appendix A).

### 3.3. Tumour-Bearing Properdin-Deficient Mice Have Dysregulated Systemic and Tumour Associated Responses

C5a, sCb-9, and CCL2 were measured at endpoint of the study (14 days) in the sera of tumour-bearing properdin-deficient and wildtype and were found to be significantly decreased in the deficient animals (Figure 2).

We next examined whether the reduction of these immune mediators in serum would mirror alterations in MDSC and M2 type macrophages of properdin-deficient mice from wildtype mice. Single cell suspensions were prepared from spleens and tumours of experimental mice 14 days after subcutaneous injection of B16F10 cells in their flanks. We first examined the frequency of MDSCs in the tumours of properdin-deficient and wildtype mice by staining with lineage-specific antibodies (Appendix A). Noticeably, properdin-deficient mice had fewer MDSCs (CD45, Gr-1, and CD11b) compared with wildtype mice (Figure 3a). The lower frequency of MDSCs in the tumours of properdin-deficient mice compared to wildtype animals was mirrored in the splenocyte population, where MDSCs and M2 macrophage numbers were reduced in tumour bearing properdin-deficient animals. Tumour-bearing properdin wildtype mice had a higher number of CD206^+^ F4/80^+^ macrophages in the spleen compared with tumour-bearing properdin-deficient mice (Figure 3b,c).

Taken together, this study shows that, whilst properdin deficiency did not alter tumour growth at 14 days post implantation, it significantly diminished inflammatory mediators in serum and tumours as well as immunosuppressive cell populations in the tumour and the periphery.

## 4. Discussion

Using B16 F10 cell-conditioned media, we have previously described that bone marrow-derived macrophages from properdin-deficient mice were skewed towards an M2 phenotype compared to bone marrow-derived macrophages from wildtype mice [13]. In our in vivo model, however, properdin-deficient mice, when inoculated with logarithmically growing B16F10 cells, showed decreased levels in blood of the chemoattractants C5a and CCL2 and in tumour of the mediators CCL2, CCL3, IL-13, and TIMP-1. There were significantly fewer M2-skewed macrophages in the spleens of tumour-bearing properdin-deficient mice.

We have previously described the complement system to exert measurable effects, acting as adjuvant mediator of anti-tumour immune surveillance, adjuvant promoter of tumour growth, and as factor in tumour immunoediting [15]. In syngeneic tumour mouse models, the general, consistent, observation is that inhibition or blockade of complement is favourable to limiting solid tumour growth over an experimental duration of 12–42 days [9,11,12,16,17].

C5a is important in the production of the chemoattractant CCL2, as in vitro and in vivo experiments have previously shown [18,19]. The origin of elevated levels of circulating CCL2 is likely to be tumour derived, as our study finds significantly elevated levels of CCL2 in tumour lysates prepared from wildtype mice. CCL2 may have autocrine and paracrine, tumour promoting, effects [20]. A mouse model of laser-induced choroidal neovascularisation demonstrated that complement activation and C5b-9 formation precedes the production of CCL2 [21]. In our study, the greater levels of C5a in tumour-bearing wildtype mice were commensurate with elevated CCL2 levels in this group. In the presence of properdin, complement activation has a greater turnover and produces enhanced levels of activation (C5a) and endproducts (C5b-9) when stimulated. CCL2 has recently been shown to be involved in tumour progression, where the genetic absence of CCL2 inhibited recruitment of splenic myeloid-derived suppressor cells immune cells and reduced tumour growth in a syngeneic model [22].

CCL2, CCL3, TIMP-1, and IL-13 are products readily found in an immune response that is driven by M2 macrophages [23], which correspond to a tumour growth-promoting phenotype that aids tumour cell proliferation, reduction of an antigen-specific immune responses, and increased angiogenesis. M2 type macrophages have been shown to induce the differentiation of regulatory T cells and co-determine the progression of numerous cancers [24]. Analysis of the systemic compartment (spleen) revealed a significant reduction of M2 type (CD206^+^) macrophages in tumour-bearing properdin-deficient mice compared to wildtype mice. We propose that properdin deficiency diverted a poorly immunogenic tumour into an “immunologically cold” one [25]. B16F10 cells do not express properdin mRNA (own data and [3]), thus it can be concluded that suppression is due to systemically circulating properdin rather than tumour-produced properdin. However, as complement has a “tonic role” in cell homoeostasis [26] and complement components and receptors are expressed by immune and non-immune cells [27], a possibility that has not been addressed in any of the studies of syngeneic tumour models is that cell activity, in addition to cell recruitment, may differ in the absence (targeted or genetic) of complement [28].

MDSCs, which we found elevated in the spleens and tumours of properdin wildtype mice, can suppress anti-tumour immunity indirectly. Like M2 tumour-associated macrophages, MDSCs generate the cytokines IL-10 and TGF-β that can suppress anti-tumour tumour-infiltrating leukocytes, generate regulatory T cells in tumour, and convert dendritic cells into a regulatory phenotype. MDSCs may also recruit T-regs to tumours in a TGF-β-independent pathway [29].

We recognize that there are limitations to this work, which is largely descriptive and employs only one *C57BL/6* syngeneic model system for evaluating the effect of properdin on tumour growth. However, our findings have important implications for the role of properdin in modulating the tumour microenvironment, which may be important for therapy. A combination approach employing properdin blockade and checkpoint blockade inhibition may be an attractive strategy and warrants further evaluation.

Taken together, our in vivo data corroborate our published in vitro data. They indicate that properdin expression may promote an immunosuppressive tumour microenvironment in part by increasing the expression of chemoattractants (C5a and CCL2) known to recruit MDSCs and M2 macrophages. Combination strategies that include targeting the properdin pathway may be useful in the immunotherapy of melanoma. It is of interest that nucleic acid aptamers have been developed for C5a as well as CCL2 and are in clinical trials [30].

## 5. Conclusions

Properdin is a significant contributor to levels of C5a and CCL2 that are produced in this murine model of melanoma and may play a role in orchestrating immunosuppressive cells in the tumour microenvironment and periphery.

## Figures and Tables

**Figure 1 medicina-57-00085-f001:**
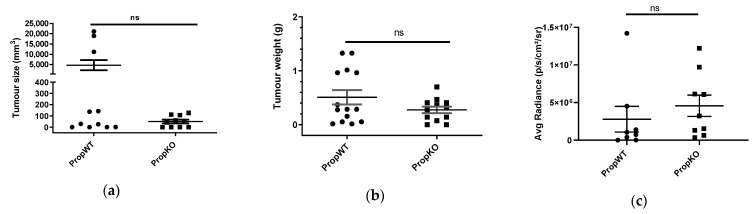
Tumour burden in properdin-deficient and wildtype mice. Wildtype (PropWT) and properdin-deficient (PropKO) mice were injected subcutaneously with B16F10 cells. After 14 days, the tumour size was measured by callipers. Values are expressed as mean ± SEM from two independent experiments (**a**). The weight of tumours was established from three independent experiments (including the ones described in a) and values are expressed as mean ± SEM (**b**). Quantitative analysis of the luciferase signal emitted from the tumour after injection of substrate 12 days following subcutaneous injection of B16F10-luc cells. Values are expressed as mean ± SEM from two independent experiments (**c**). Data were analysed by two-tailed Mann–Whitney test.

**Figure 2 medicina-57-00085-f002:**
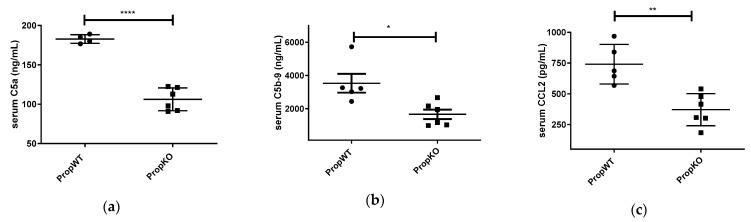
Serum levels of properdin- deficient (PropKO) and wildtype (PropWT) tumour-bearing mice for levels of C5a (**a**), C5b-9 (**b**), and CCL2 (**c**). Data are presented as means ± SEM. Two tailed *t*-tests were performed, * *p* < 0.05, ** *p* < 0.01, **** *p* < 0.0001.

**Figure 3 medicina-57-00085-f003:**
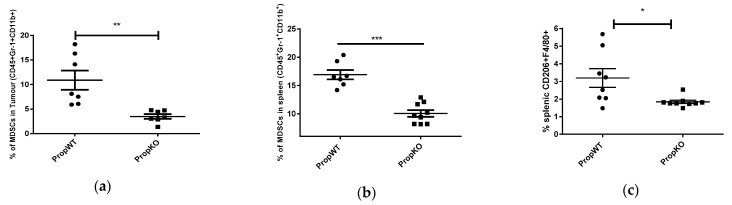
CD45^+^ CD11b^+^ Gr-1^+^ population in tumours and spleens and CD206^+^F4/80^+^ population in spleens from properdin-deficient (PropKO) and wildtype (PropWT) mice. Tumour cell suspensions were analysed for the proportion of CD45^+^ CD11b^+^ Gr-1^+^ cells (**a**) and splenic suspensions for the proportions of CD45^+^ CD11b^+^ Gr-1^+^ cells (**b**) and of CD206^+^F4/80+ cells (**c**). The data are presented as means ± SEM. Statistical analysis was performed by unpaired *t*-test. * *p* < 0.05, ** *p* < 0.01, *** *p* < 0.001.

**Table 1 medicina-57-00085-t001:** Identification of genotype-discriminatory mediators by Mouse Cytokine Array in tumour lysates of wildtype (PropWT) and properdin-deficient (PropKO) mice.

	PropWT (*n* = 3)	PropKO (*n* = 3)	*p*-Value
TIMP1	45.72 ± 4.19	10.41 ± 0.91	<0.002
IL-13	5.24 ± 1.36	0.88 ± 0.35	<0.05
CCL2	97.49 ± 1.11	1.37 ± 0.17	<0.0001
CCL3	46.31 ± 3.01	0.93 ± 0.41	<0.0002

Three tumours of comparable sizes from each genotype were pooled, and extracted protein was analysed using Mouse Cytokine Array (R&D Systems). Reactivities were quantified densitometrically. Data are expressed as means ± *SD* and were analysed by unpaired two-tailed *t*-test.

## Data Availability

Data will be shared on request and is contained in PhD thesis Al-Rayahi, I. A.M. 2017. Role of properdin in tumour growth and cell recruitment. University of Leicester. ISNI 0000 0004 6348 7063.

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
