# Peer review of "Properdin Is a Modulator of Tumour Immunity in a Syngeneic Mouse Melanoma Model"

_medicina, 2021, doi:10.3390/medicina57020085_

Round 1
Reviewer 1 Report
I am satisfied by the modifications and corrections made by the authors.
Author Response
We thank you for the guidance you provided.
Reviewer 2 Report
Although the authors respond to all questions/concerns there are some remaining gaps.
1st the Figure S1 is not mentioned and described in the text. Also the figure legend of S1 is not appropriate. What do you really see? Is this really the gating strategy? I guess not. If the authors do not know what a gating strategy is or looks like, how did they come up with their results?
2nd it is acceptable not to do the recommended stainings however there are ways to stain heavily pigmented B16 tumors. The authors should increase their capabilities in this respect.
3rd the array data are still not nicely presented. What are the values of table 1? Are they normalized somehow, if not why? I recommended to do the analysis as recommended by R&DSystems (https://www.rndsystems.com/products/proteome-profiler-mouse-cytokine-array-kit-panel-a_ary006)
but why did they not follow this hint and did not prepare a corresponding diagram (see bar digrams and compare units)?
Author Response
Please see hereafter our replies.
Although the authors respond to all questions/concerns there are some remaining gaps.
We thank reviewer 2 for their additional helpful comments to improve the manuscript which we hope are now addressed to your satisfaction.
1st the Figure S1 is not mentioned and described in the text.
Corrected (line 170-171)
Also the figure legend of S1 is not appropriate. What do you really see? Is this really the gating strategy? I guess not. If the authors do not know what a gating strategy is or looks like, how did they come up with their results?
Please see the additional plots demonstrating the gating strategy employed. Please note that when these experiments were done we were not able to do doublet discrimination and viability stains in combination with our lineage markers. We have gated on FSC/SSC parameters and then on CD45 high cells.
2nd it is acceptable not to do the recommended stainings however there are ways to stain heavily pigmented B16 tumors. The authors should increase their capabilities in this respect.
We agree with reviewer 2 that this is an area in which we should improve our capabilities.
3rd the array data are still not nicely presented. What are the values of table 1? Are they normalized somehow, if not why? I recommended to do the analysis as recommended by R&DSystems (https://www.rndsystems.com/products/proteome-profiler-mouse-cytokine-array-kit-panel-a_ary006)
We have done the recommended analysis (detailed in methods 2.4) and have added additional normalised pixel data for the 6 cytokines that were expressed (supp. fig 2).
but why did they not follow this hint and did not prepare a corresponding diagram (see bar digrams and compare units)?
We have now shown bar diagrams similar to the R&D website but only for the 6 cytokines that came up on the array to improve manuscript clarity.
Reviewer 3 Report
The authors have satisfactorily addressed the concerns.
Author Response
We would like to thank you for the guidance you provided.
This manuscript is a resubmission of an earlier submission. The following is a list of the peer review reports and author responses from that submission.
Round 1
Reviewer 1 Report
In their manuscript entitled “Properdin is a modulator of tumour immunity in a syngeneic mouse melanoma model” Rayahi and co-workers investigate the contribution of the complement system to tumour immunity. In particular, they studied the role of properdin, a protein that augments complement activation by comparing the response of wild type and congenic properdin knockout mice to subcutaneous injections of syngeneic B16F10 melanoma cells. They found no difference in tumour growth, but properdin-deficient mice had fewer intratumour and splenic myeloid derived suppressor cells (MDSC). These mice also showed reduced M2 macrophages and reduced serum complement C5a, sC5b-9 and CCL2 chemokine. Based on these observations, they authors hypothesise that the intact complement system in the wild type mice lessens tumour immunity.
In honesty, this feels like a pilot study wherein the authors establish their methodologies as part of a larger study. The strength of this paper is that the study is logically well laid out and rigorously conducted. The major drawback lies in its descriptive nature. Though limited, this manuscript is self-contained and well presented. Within its limited scope, the authors are encouraged to address the following comments/questions:
1) Why did the wild type mouse show greater variability in tumour growth?
2) The significant immunological phenotype in response to B16F10 in properdin KO mice would indicate a difference in tumour immunity between the two groups. They authors should test this. Would injecting less cells accentuate the differences in tumour immunity between the wild type and properdin-deficient mice?
3) Have the authors experimented with another C57BL/6-derived syngeneic cell model?
Minor comments:
- lines 30-32 - These sentences have punctuation errors.
- line 45 - A new paragraph?
- lines 130-1 - the B16-Luc line expresses luciferase, not luciferin.
Author Response
Whilst we agree with the reviewer that the study is somewhat descriptive we disagree that this work represents a pilot study as it required two project licenses with amendments to complete this work. However, we have indicated the limitations of the project in the discussion section (line 232-236). Furthermore, we have been careful not to over-interpret our results and have reduced speculation to a minimum.
ad 1)
We agree with the reviewer’s astute observation that there is more variability in tumour growth in wild type animals. In fact, tumour growth appears to be somewhat biphasic which we are unable to explain but may underline a tumour promoting role of properdin where more uncontrolled outgrowth of tumour is permitted. However, knockout mice clearly have reduced tumour size.
ad 2)
We have assessed tumour growth in syngeneic mice using our two related cell lines (B16F10-Luc and B16F10) and the cell numbers employed were based on minimum cell numbers below which no tumours could be established and are comparable with similar studies (Brentville et al. 2019). This model, unlike others is an orthotopic model and could be studied only up to a diameter of tumour of 1 cm, which is sooner than other studies conducted elsewhere.
We agree that it would be interesting to alter input cell numbers injected into the animals however this is beyond the scope of this project. We have aimed to maintain the principles of the 3Rs reducing use of experimental animals to a minimum.
ad 3)
No- although this would be important to do for future work to confirm the validity of these findings across experimental models. We have added a sentence (line 233) in the discussion to highlight this.
The minor corrections have been done (lines 30-32 / 45/ 130-131).
Reviewer 2 Report
The authors present a B16 melanoma mouse model in the context of properdin ko mice or wildtype animals. Tumor growth was not influenced by the ko of properdin, however lower numbers of MDSC were found in the corresponding spleens and tumors. Further characterisation of the macrophages and cytokine measurements revealed some intriguing differences.
The following points should be addressed:
- A gating strategy plus examples of the flow cytometry experiments should be presented (CD45+CD11b+ Gr-1+; CD206+ F4/80+detection)
- IHC of tumors or spleens for macrophages would be nice to see. Are there differences in total numbers?
- Please provide the results (chemiluminescent detection of the array with quantification diagram--> compare R&D webpage)
Author Response
Thank you for your appraisal.
In the revised manuscript we have provided example plots in the supplementary results (Sup. Figure 1).
We are unable to provide this data (IHC macrophages) - In our hands this was hard to evaluate because of the pigmentation of the tumour cells complicating analysis.
We have provided an example diagram of the arrays in the supplementary results (Sup. Figure 2).
Reviewer 3 Report
The manuscript titled “Properdin is a modulator of tumour immunity in a syngeneic mouse melanoma model” by Al Rayahi et al., evaluated whether properdin is an important modulator of tumour growth in Vivo in mouse melanoma model using the WT and Properdin deficient mice.
Their result suggests that tumors of Properdin deficient mice has lower chemokine, MDSCs and macrophages.
I do have quiet a few concerns about the study:
- What is the novelty of this study? What is the biological and clinical implication of this study? Properdin is already known to have effects on Innate immune system. How does these data/findings add to the already known facts?
- If properdin can act as an immunomodulator, it should enhance the immune mediated response. The authors should test whether the response to Anti PD1 or Anti PDL1 drugs is higher in properdin deficient mice compared to WT counterparts.
- The main findings of the study is that the tumors of Properdin deficient mice has lower chemokine, MDSCs and macrophages. The result has been shown using only B16F10 cells. The B16F10 cells has their own drawbacks. These results should be validated using another line such as any of the Yumm lines or Yummer lines.
- Does the properdin knockout mice exhibit any other phenotypes that distinguishes them from their WT counterpart? Does the coat color or tail/paw color is different in the knockout than the WT?
Author Response
ad 1. As the reviewer helpfully pointed out, properdin can act as an immune modulator. Our previous in vitro and in vivo work has shown that the activity of complement properdin / intact alternative pathway of complement extends beyond the humoral phase and influences cellular function. Contrasting with other complement knockouts that have been used, e.g. C3 knockouts, this model does not completely abrogate the production of complement split products, but rather reduces them. Properdin is the only positive amplifier of complement activation, so use of this model allows us to investigate fine tuning of the system in a variety of contexts including cancer.
We have demonstrated that the TME is altered in properdin deficient animals. This work suggests that properdin blockade in combination with other therapies (i.e. checkpoint blockade inhibition) may provide a rational approach to therapy although this would be the subject of future work. We have clarified the novelty of the project in the discussion section (lines 232-236).
ad 2.
We agree with the reviewer that these are interesting and important experiments to do but are beyond the scope of this project. We have added text (line 235-236) to highlight this point.
ad 3.
We agree that it would be interesting to repeat these experiments with another cell line. However, the B16F10 cell line model is a highly validated experimental model system and testing a panel of other cell lines is beyond the scope of this project. We have added text (page 232-233) to indicate this limitation. We have also aimed to maintain the principles of the 3Rs, reducing use of experimental animals to a minimum whilst deriving important biological information.
ad 4.
Our review of properdin mice is detailed in http://www.researchtrends.net/tia/article_pdf.asp?in=0&vn=19&tid=36&aid=6229 and compares baseline wild type and knockout phenotypes.